# Absence of 2899C<T Mutation in the WNK4 Gene in a Free-Ranging Lion (*Panthera leo*) with Polymyopathy

**DOI:** 10.3390/ani12030389

**Published:** 2022-02-08

**Authors:** Desiré L. Dalton, Chantelle Pretorius, Lin-Mari de Klerk-Lorist, Bjorn Reininghaus, Peter Buss, Emily P. Mitchell

**Affiliations:** 1South African National Biodiversity Institute, P.O. Box 754, Pretoria 0001, South Africa; D.Dalton@sanbi.org.za (D.L.D.); C.Pretorius@sanbi.org.za (C.P.); 2Skukuza State Veterinary Office & Laboratory, Directorate Animal Health, Department of Agriculture, Land Reform and Rural Development, Kruger National Park, P.O. Box 12, Skukuza 1350, South Africa; LinmarieDK@dalrrd.gov.za; 3Mpumulanga Veterinary Services, Thulamahashe, P/Bag X11309, Mbombela 1200, South Africa; bjorn.reininghaus@gmail.com; 4Veterinary Wildlife Services, South African National Parks, P.O. Box 86, Skukuza 1350, South Africa; peter.buss@sanparks.org; 5Department of Paraclinical Sciences, Faculty of Veterinary Science, University of Pretoria, P/Bag X01, Onderstepoort 0110, South Africa; 6Centre for Veterinary Wildlife Research, Faculty of Veterinary Science, University of Pretoria, P/Bag X01, Onderstepoort 0110, South Africa

**Keywords:** polymyopathy, WNK4, lion, *Panthera leo*

## Abstract

**Simple Summary:**

Samples from an African lion cub in the Greater Kruger National Park area (South Africa), which could not walk, were tested for a gene mutation that causes one type of muscle weakness in domestic cats. The cause of the muscle weakness is believed to be genetic, but our study showed that the mutation that is found in similarly affected domestic cats was not present in the cub. Genetic diseases are more common in inbred animal populations, so this condition needs to be further evaluated to assist in the conservation of these magnificent creatures.

**Abstract:**

Polyphasic skeletal muscle degeneration, necrosis and mineralization of skeletal muscle was diagnosed in eight juvenile free-ranging lions (*Panthera leo*), from five different litters in the Greater Kruger National Park area that were unable to walk properly. A detailed investigation was not possible in free-ranging lions, so the cause could not be determined. The cases resembled hypokalemic polymyopathy in domestic cats with muscle weakness. A candidate-gene approach previously identified a nonsense mutation in the gene coding for the enzyme lysine-deficient 4 protein kinase (WNK4) associated with the disease in Burmese and Tonkinese cats. In this study, we sequenced all 19 exons of the gene in one case, and two control samples, to identify possible mutations that may be associated with polymyopathy in free-ranging lions. Here, no mutations were detected in any of the exons sequenced. Our findings indicate that the WNK4 gene is not a major contributor to the condition in these lions. Further studies into the pathogenesis of this condition are needed to inform conservation policies for this vulnerable, iconic African species.

## 1. Introduction

Severe skeletal muscle weakness resulting in difficulties in standing and walking was observed in eight free-ranging lion (*Panthera leo*) cubs (<10 months old) from five separate litters in several locations in the Greater Kruger National Park area (KNP), South Africa, between 2015 and 2020. The cub number, date of death, litter, age and sex of the lion cubs are shown in Table 1.

Cubs 1–3 and 5 were found dead; the remaining cubs were euthanized. Cubs 7 and 8 had been abandoned by the pride. The cubs were variously described as having paralyzed front legs (Cubs 3 and 4), crawling (Cub 6) and reluctant to walk, finding it difficult to stand (especially on the hind limbs), and creeping along with all four limbs flexed (Cubs 7 and 8). No traumatic lesions were found; the cubs had reduced skeletal muscle mass. Full post-mortem examinations (after euthanasia) of the cases showed multiple pale tan tracts (1–5 cm dia) in many skeletal muscle masses including the semi-membranous, pelvic girdle, intercostal, occipital, masseter and periocular muscles (Figure 1).

The histological changes in each case were similar: severe polyphasic skeletal muscle fiber degeneration, necrosis, regeneration and mineralization. Scattered individual or clusters of myofibers were variably swollen, hypereosinophilic and fragmented with variable loss of striations, and karyolysis or large multiple nuclei in rows interpreted as regeneration (Figure 2). Satellite cell hypertrophy and a few myofibers contained deeply basophilic particulate material interpreted as dystrophic mineralization.

The cause of the clinical signs and lesions could not be determined. Detailed clinical examination and serum biochemistry were not possible in these cases because the animals were found dead or, in the case of live animals, park policy is to only examine and treat human-made injuries such as those caused by snares or vehicular accidents. In addition, the cubs died in locations over 400 km from a veterinary laboratory.

Polymyopathy may have many causes in domestic cats [1]. Muscle fibers were not of varying sizes, myofibers were not hyperplastic and did not show splitting and endomysial fibrosis were absent. Therefore, muscular dystrophy caused by alpha-dystroglycan, beta-sarcoglycan and laminin deficiencies and congenital myotonia were thought to be unlikely, but could not be ruled out. Three cubs were female, ruling out X-linked muscle dystrophy. No evidence of glycogen storage disease was present in any tissues. Frozen muscle sections were not available to check for congenital nemaline myopathy, although the myofiber atrophy seen in this condition was not present. Lesions compatible with toxoplasmosis were not seen in the skeletal muscle, or in any other organ. Vitamin E and/or selenium deficiency are uncommon in carnivores and were also thought to be unlikely in free-ranging lions. Toxic myopathies were thought to have rather presented as outbreaks in lions of varying ages, whereas in this series only cubs were affected.

Skeletal muscle weakness has also been observed in two South-African-born six-month-old white lion cubs housed in a North American zoo [2]; in these animals, extensive clinical diagnostic investigation suggested that the clinical syndrome was similar to hypokalemic polymyopathy in domestic cats. This possibility was explored in this study.

Hypokalemic periodic paralysis in humans is a rare disorder that was first described by Musgrave in 1972. The disorder is usually diagnosed in late childhood or in the teenage years [3]. The condition results in periodic extreme muscle weakness, which results in an inability to move muscles in both the arms and legs lasting from hours to days. Triggers include a cold environment, stress, fasting, rest after exercise, a viral illness, certain medications and/or pregnancy. The severity of attacks varies between individuals and can occur daily, weekly, monthly or only rarely [4]. Both inherited and acquired causes of hypokalemic periodic paralysis have been identified. Mutations identified in the Calcium Voltage-Gated Channel Subunit Alpha1 S (CACNA1S), Sodium Voltage-Gated Channel Alpha Subunit 4 (SCN4A) and Potassium Inwardly Rectifying Channel Subfamily J Member 2 (KCNJ2) that encode the voltage-gated channels in muscle membranes that generate membrane potentials has been found associated with the disorder in humans [5]. Furthermore, missense mutations in Ryanodine receptor type 1 (RYR1), which promotes release of calcium within myofibrils, were recently identified in patients with periodic paralysis [6]. In addition, acquired hypokalemic periodic paralysis has been associated with thyrotoxicosis (high-circulating blood thyroid hormone levels).

In animals, hypokalemia resulting in skeletal muscle weakness was first reported in domestic cats (*Felis catus*) by Eger et al. [7]. Since then, it has been reported in several countries and has been shown to be due to an autosomal recessive trait that affects young Burmese and Tonkinese cats. Affected kittens display varying degrees of weakness in the neck, thoracic limb girdle and appendicular muscles [8]. Thus far, the disease has been linked to a single nonsense mutation, producing a premature stop codon in the serine-threonine kinase gene (‘with no kinase 4′, WNK4) that codes for lysine-deficient 4 protein kinase [9]. The enzyme is expressed mainly in the kidney, while the pancreas, bile duct, brain, epididymis and skin show lower enzyme levels [9,10,11]. WNK4 is involved in the regulation of complex pathways in the renal distal convoluted tubule, that in turn regulate the activity of all major sodium and potassium transporters [12]. Thus, altered function in WNK4 causes loss of potassium in the urine in kittens, which in turn results in symptomatic hypokalemia. Potassium is necessary for the muscle contraction, so hypokalemia results in muscle weakness.

This project sought to investigate if the 2899C<T mutation of the WNK4 gene is present in African lions, to determine whether or not the polymyopathy syndrome in free-ranging lions could be due to hypokalemic polymyopathy.

## 2. Materials and Methods

This study included one case sample from an affected free-ranging lion (Cub 6) based on necropsy and histological examination, and two unaffected lion samples stored in the SANBI Biobank as controls. Genomic DNA was extracted from tissue samples using the Quick-DNA Miniprep Plus Kit (Zymo Research, Irvine, CA, USA) following the manufacturer’s protocol. In order to determine concentration and purity of the extracted DNA, a Nanodrop ND-1000 Spectrophotometer (Inqaba Biotech, Pretoria, South Africa) was used. Amplification was conducted using fourteen published WNK4 primer sets developed by Gandolfi et al. [8]. The primers used the published human WNK4 exons (GenBank number: NM_032387.4) to identify cat exonic sequences in the available cat genome assembly. Primers developed from domestic cat are often used to genotype other feline species, including lions [13,14]. Thus, this published primer set was used in this study to amplify all 19 exon regions of the WNK4 gene in lion. All Polymerase Chain Reaction (PCR) cycling was conducted in a MiniAmp Plus (Thermo Fischer Scientific, Carlsbad, CA, USA) thermocycler in a final reaction volume of 15 µL containing 6.25 µL Ampliqon *Taq* DNA Polymerase Master Mix RED (Ampliqon, Odense, Denmark); 0.5 µL of the forward and reverse primers (Thermo Fischer Scientific, Carlsbad, CA, USA); 5.25 µL double distilled water (ddH_2_O) and 2.5 µL of the template DNA. The PCR amplification conditions were as follows: one cycle at 95 °C for 5 min followed by 35 cycles at 95 °C for 30 s, 55–62 °C for 30 s and 72 °C for 30 s; a final extension step of one cycle at 72 °C for 10 min. The PCR product was then run on a 2% agarose gel to ensure that amplification was successful and then purified by adding 0.25 µL Exonuclease 1 and 1 µL FastAP enzymes (Thermo Fisher Scientific, Carlsbad, CA, USA) to the PCR product. The purification reaction was run for one cycle at 37 °C for 15 min followed by one cycle at 85 °C for 15 min in a MiniAmp Plus (Thermo Fischer Scientific, Carlsbad, CA, USA). Cycle sequencing reactions were completed using the BigDye Terminator v3.1 Cycle Sequencing Kit (Thermo Fisher Scientific, Carlsbad, CA, USA) using the Sanger chain termination method. Sequencing was conducted in a final reaction volume of 10 µL containing 0.7 µL BigDye, 2.55 µL of BigDye Terminator v3.1.5 × Sequencing buffer, 0.75 µL double distilled water (ddH_2_O), 1 µL of primer and 5 µL of the PCR product. Sequencing reactions were completed for both the forward and reverse direction with the cycling conditions as follows: one cycle at 94 °C for 2 min; 40 cycles at 85 °C for 10 s, 53 °C for 10 s and 60 °C for 2 min 30 s. The cycle sequencing product was purified using the BigDye XTerminator Purification Kit (Thermo Fisher Scientific, Carlsbad, CA, USA) and sequences were visualized using the 3500 Genetic Analyzer (Thermo Fisher Scientific, Carlsbad, CA, USA). Forward and reverse were aligned to create a consensus sequence and all sequences were manually trimmed in BioEdit [15]. Additionally, sequences were aligned in MEGA7 [16]. Lastly, sequences were visually inspected checked for ambiguous peaks in BioEdit [15]. All sequence reads were obtained using standard Sanger sequencing methods and resulting sequence traces were submitted to nucleotide blast and were blasted against the Coding Sequence (CDS) of WNK4 from Burmese cats (GenBank accession number: JQ522971), in order to ensure that the correct region of WNK4 was obtained. Accession details for DNA sequences used in the study have been submitted to Genbank (GenBank accession numbers: OK513187 and OK513188).

## 3. Results

A mutation (c.2899C>T) is reported to cause a premature stop codon (CAG > TAG) in Burmese cats with hypokalemic polymyopathy (Figure 3A). WNK4 sequences were successfully amplified for all primers sets tested, indicating that this gene is highly conserved. To examine a possible association between alterations in the WKN4 gene and susceptibility to polymyopathy syndrome in a free-ranging lion cub, we sequenced all 19 exons in three samples (one case and two controls). Alignment of sequences in MEGA7 did not identify any nucleotide differences between the case or control samples. In addition, visual inspection did not detect any ambiguous peaks. Thus, genotype analysis demonstrated that the c.2899C>T mutation was not present in either the case or control samples (Figure 3B). Further, mutations were not found in any of the exons sequenced in the coding regions of the WKN4 gene in either case or control samples.

## 4. Discussion

The diagnosis of the cause of polymyopathy in these lion cubs remains uncertain, given the limited investigation that was possible in free-ranging lion cubs. The sporadic presence of the syndrome and absence of clinical signs in adult animals is strongly suggestive of a genetic cause with recessive inheritance and/or incomplete penetration. It is possible that the polymyopathy syndrome is multi-factorial and contributing factors may include both environmental and genetic factors as described in humans with hypokalemic periodic paralysis [4].

We found no evidence of mutations in the WNK4 gene, but cannot rule out the possibility of altered WNK4 gene expression. In South Hampshire sheep with neuronal ceroid lipofuscinoses (Batten disease), a causative mutation in the Ceroid Lipofuscinosis Neuronal 6 (CLN6) gene was absent; however, quantitative real-time PCR revealed that CLN6 mRNA expression was one-third lower in affected South Hampshire sheep compared to control sheep [17]. The authors attributed this finding to an unidentified mutation in or near the CLN6 gene resulting in lower mRNA expression. Thus, although the WNK4 gene mutation identified in domestic cats was absent in the lions, other alterations in or near the gene may result in downregulation of the transcription of the gene or mRNA stability resulting in hypokalemia and polymyopathy.

An alternative hypothesis is that mutations in other genes may be involved in the pathogenesis of the polymyopathy syndrome. Mutations in the CACNA1S, RYR1, SCN4A gene have been reported to be associated with hypokalemic periodic paralysis (HOKPP) in humans [5]. In addition, mutations in KCNJ2 have been found associated with non-familial HOKPP, and with thyrotoxic periodic paralysis in humans [18]. Although polymorphisms in these genes have not been found to be associated with hypokalemia in domestic cats [19], their involvement in this polymyopathy syndrome in free-ranging lions cannot be currently be ruled out.

This study only examined one animal with polymyopathy. A genome-wide association study with a large number of case and control samples may identify further possible genes associated with the disorder in the future. This technique refers to the screening of markers of genetic variation known as single-nucleotide polymorphisms (SNPs) distributed throughout the genome in cases and controls to identify susceptibility regions for complex diseases. This method has been used in the analysis of several human diseases and is currently available for several domestic species such as cattle (*Bos taurus*), dogs (*Canis lupus familiaris*), sheep (*Ovis* sp.), pigs (*Sus* sp.), horses (*Equus caballus*) and chickens (*Gallus gallus domesticus*). Although this is a recently established method in domestic animals, it has been successfully used to identify disease-causing genes as well as genetic mechanisms of quantitative traits [20]. The disadvantages of this method are that it requires a reference genome sequence where SNPs have been mapped on the genome; many cases and controls must be tested and that it only detects a region that is associated with disease. Further analysis would be needed to identify the exact SNP involved in the disease. Due to these limitations, little to no studies have used this method in wildlife species.

Additional methods to identify variants associated with inherited diseases in non-model organisms include whole-genome sequencing (WGS) or whole-exome sequencing. These technologies are becoming cheaper, easier and can provide large volumes of data [21]. Whole-genome sequencing refers to a comprehensive analysis of the entire genome and has been used to identify gene mutations associated with early-onset progressive retinal atrophy in African black-footed cats (*Felis nigripes*) [22]. Whole-exome sequencing, on the other hand, is more cost effective than WGS and involves sequencing the complete set of protein-coding regions (exons). This method has identified deletions in the sarcoglycan genes in Boston terriers that were found to be associated with limb-girdle muscular dystrophy [23].

Interactions between genetic and environmental effects can be complex and may result in variation in expression of the disease phenotype. In addition, the disease may be polygenic, thus an as-yet-unidentified gene or combination of genes may be responsible for polymyopathy in lions. Novel technologies such as next-generation sequencing should make it possible to perform case/control studies that could eventually shed more light on the etiology of polymyopathy. However, sample sizes would need to be increased and pedigree analysis conducted, which present significant challenges in free-ranging lion populations. These studies may be easier to perform in captive lions.

Low genetic diversity leading to inbreeding is associated with higher mortality, lower fecundity, developmental instability, more frequent developmental defects and greater susceptibility to disease [24]. For example, Burmese and Tonkinese cat breeds, which are particularly susceptible to hypokalemic polymyopathy, have low heterozygosity and high inbreeding coefficients [25]. Throughout their range, lion populations have declined by more than 42% over the past 21 years, coinciding with the rise of a European colonial presence [26], which has led to increased rates of inbreeding in southern African lions [27]. Inbreeding is further exacerbated in lion populations in South Africa that are confined to small, enclosed reserves [28]. Although reduced genetic diversity may be present in southern African lions, the WNK4 gene appears not to be affected (at least in this small number of cases).

## 5. Conclusions

These cases highlight the inherent difficulties in investigating and documenting disease in free-ranging felids. No mutations were found in the coding regions of the WNK4 gene in a free-ranging lion cub from KNP with polymyopathy. Further investigation is needed to determine serum potassium and creatinine kinase levels in affected cubs and whether or not the condition in fact has a genetic basis. Although, non-genetic nutritional, toxic and metabolic causes of polymyopathy are unlikely in free-ranging animals, and typically affect animals of varying ages, such causes need to be additionally considered. This study therefore supports further investigation into polymyopathy in free-ranging African lion populations. The worldwide prevalence of this disorder should be determined, particularly in captive lions, which can be more easily evaluated than their free-ranging counterparts.

## Figures and Tables

**Figure 1 animals-12-00389-f001:**
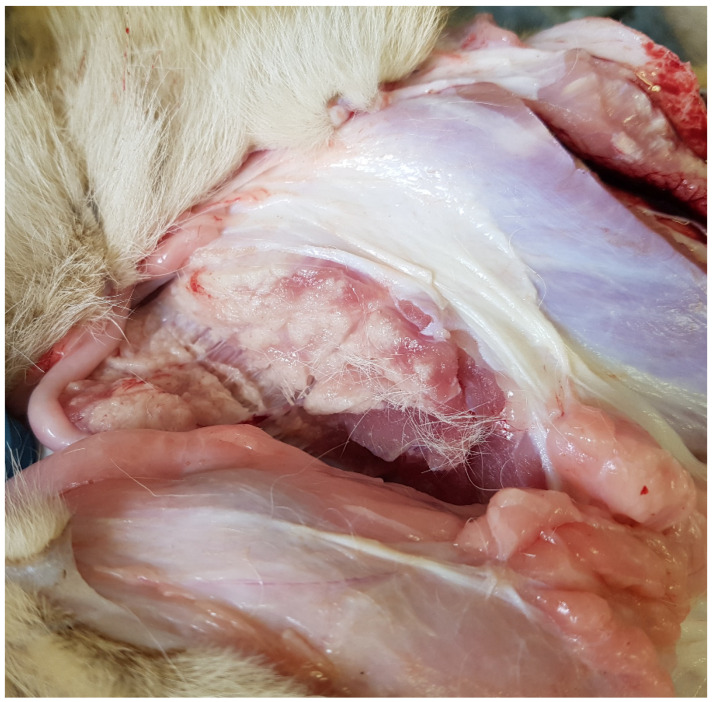
Multiple tracts of white discoloration in a cross-section of the semi-membranous muscle of Cub 6.

**Figure 2 animals-12-00389-f002:**
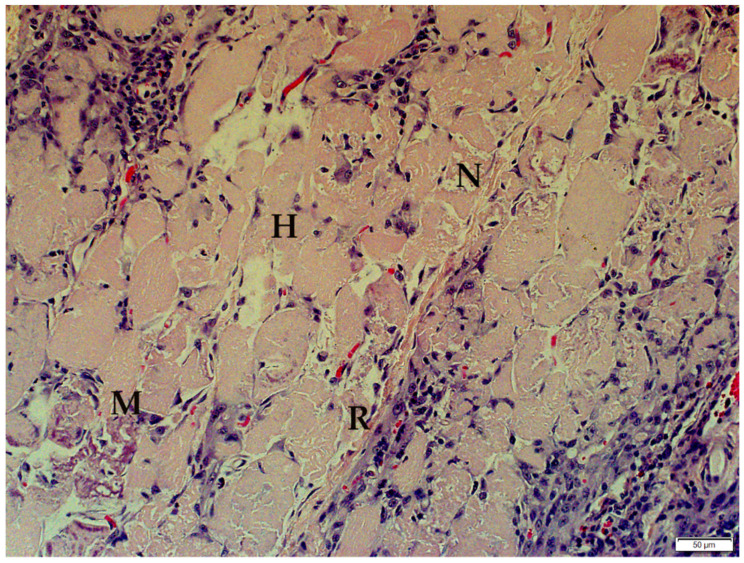
Polyphasic myopathy in Cub 8. Myofiber necrosis (N) is characterized by swollen fragmented fibers without striations. Mild mineralization (M), satellite cell hypertrophy (H) and myofiber regeneration (R) are present (Haematoxylin and Eosin stain).

**Figure 3 animals-12-00389-f003:**
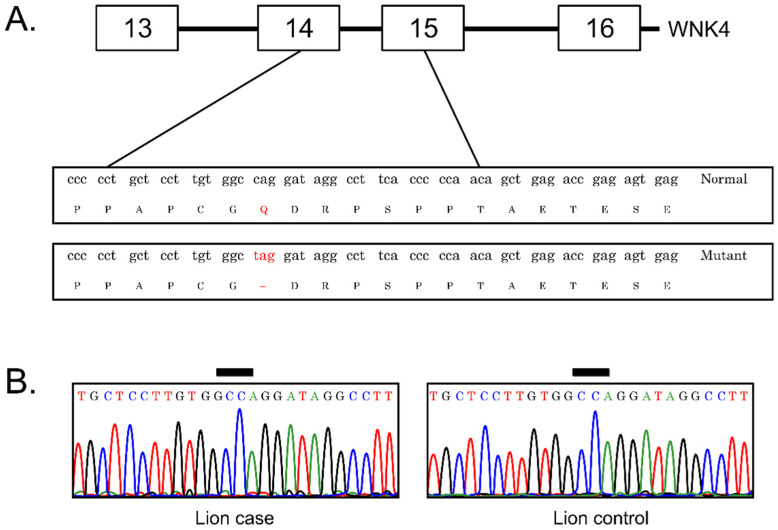
(**A**) Depiction of the predicted alteration in the WNK4 gene for the c.2899C>T mutation, which results in a premature stop codon; (**B**) representative chromatograms generated by fluorescent dye-primer sequencing of PCR products. The DNA sequence from the lion case was identical with that from healthy controls.

**Table 1 animals-12-00389-t001:** Date of death, litter, age and sex of eight cubs with polymyopathy.

Cub	Date of Death	Litter	Age ^1^	Sex
1	6 January 2015	1	Juvenile	Female
2	6 January 2015	1	Juvenile	Male
3	6 January 2015	1	Juvenile	Male
4	27 October 2015	2	2 months	Male
5	15 March 2017	3	Juvenile	Male
6	30 July 2018	4	Juvenile	Female
7	9 October 2020	5	6 months	Female
8	3 November 2020	5	10 months	Female

^1^ Exact ages not known.

## Data Availability

Not applicable. No datasets were generated or archived during this study.

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
