# Peer review of "Absence of 2899C<T Mutation in the WNK4 Gene in a Free-Ranging Lion (Panthera leo) with Polymyopathy"

_animals, 2022, doi:10.3390/ani12030389_

Round 1

Reviewer 1 Report

The authors describe a case of polyphasic myopathy in an African lion and detail the absence of mutation in a specific gene linked to one such myopathy in domestic cats. The methodology appears sound, though the primer sequences should be given- I would like to know why the primer design was based on the human WNK4 gene and not that of a more closely related felid (e.g. tiger) or the lion itself (The tiger genome and comparative analysis with lion and snow leopard genomes (nih.gov)). This is particularly odd as the sequences were BLASTed against the coding sequence for Burmese cats. 

My main concerns with this paper is the scope of the work conducted, the lack of any definitive diagnosis and the sparse data presented:

  1. It is not clear why hypokalaemic myopathy is the main differential under investigation- no clinical or clinico-pathological data are presented to suggest that this is likely. Additionally, no circumstantial genetic data (e.g. suspicion of close genetic relatedness of cases) are presented to suggest that the disease is genetic- nutritional myopathy would present very similarly and is much better described in this species.
  2. I am surprised that the clinical history and post mortem and histopathological results are not presented but for the briefest of descriptions in the introduction. Have these been published already elsewhere? A more full report of the case would allow readers to assess the significance of the negative genetic testing. 
  3. The single figure does not add much to the paper, as it shows a negative result. Images of the histology and gross pathology would be more instructive. 
  4. Discussion of the negative result mostly concerns the possibility of other mutations or downregulations that could cause hypokalaemia, with only the last line admitting that many other diagnoses need investigation as well. I worry that this paper might give the casual reader the impression that there is much more evidence for hypokalaemic myopathy than there is. 

The findings in this paper seem to build heavily on the findings from the captive cubs, for which greater work up seems to have been done (reference 11 Milnes et al.). It seems backwards for the genetic investigation to be published before the clinical and pathological findings that inspired them, so I cannot recommend this paper for publication at this time. However, the methodology is sound and I am confident that these findings will be published, either following the conference authors case report or in collaboration with them, particularly if it can be arranged for their samples to also be tested. 

Reviewer 2 Report

Review of Absence of 2899C in the WNK4 gene in a free ranging lion with polymyopathy

This is a nice paper with a novel approach towards conservation medicine and wildlife diseases.  The design of the study and presentation of the results are clear and concise, however there are a few issues.  The most important is you do not detail how you ruled out other causes of polyphasic myopathy such as vitamin E / selenium deficiencies, toxoplasmosis, etc.  Were littermates of the affected cubs otherwise normal?  Are you just trying to rule out one of many possibilities?  In te conclusion you mention that the disease might be multifactorial so it is OK if you cannot rule out other contributing factors, but some justification is needed.

Other areas of clarification:

Lines 56-60: while you provide a nice description of the clinical disease in domestic cat species, you do not provide the histologic lesion (minimal to mild polyphasic myopathy mostly in the diaphragm and intercostal muscle Jubb Kennedy and Palmer’s Pathology of Domestic Animals).  The difference between your cases in lions and that in domestic cats is important and helps further your case that there is a difference in underlying pathogenesis.

Lines 149-156: I had trouble following the logic of the sheep study and how these studies relate to your study.  I think you are trying to say there may be another of decreased WNK4 expression other than DNA sequence mechanism (such as differences in levels of mRNA expression or stability), but this link is not clearly explained.

Lines 158-161: you start two sentences out of the first three with “In addition,” which distracts from the actual information Just change one to “furthermore” or similar

Lines 177-181: For readers less familiar with GWAS, WGS and WES, please provide a brief summary of how these techniques are different from each other.  These explanations are especially important as the use of these techniques is one of the most interesting facets of this manuscript.

Reviewer 3 Report

This is a short communication describing an interesting polymyopathy affecting free ranging African lions. Unfortunately, this myopathy is not clinically characterized and the pathological changes in the muscle obtained at necropsy were basically ignored or misinterpreted. A differential diagnosis based on the histpathology would lead the authors in a different diagnostic direction. The main differential diagnosis should be a form of muscular dystrophy based on hstopathology. It is critical to know if the affected animals are all male and if the CK activity is markedly elevated. While the polymyopathy is of interest, there is too much time spent on mutations causing hypokalemic myopathy in other species and too much focus on what this likely isn’t.

Specific points:

  1. What do you mean by “unable to walk properly”? Was the gait stiff and stilted? Could the animals only do a slow walk and not run? Something further could be said about the gait just by observation. Were the affected animals all male? Is a video available? Any there any digital images of animals?
  2. “Polyphasic skeletal muscle degeneration, regeneration and mineralization” is not a diagnosis. These are pathological changes that are consistent with a degenerative and regenerative polymyopathy.  An important differential based on these changes is a form of muscular dystrophy which they do not even mention. There are many causes of muscle weakness and to focus in on just hypokalemic polymyopathy is not realistic.

Introduction. The whole focus is on hypokalemic periodic paralysis in other species. Is the episodic weakness of this condition similar to that of the clinical presentation in the lions? From the description of the gait it doesn’t sound like this is episodic but more likely continuous…

Line 65. Were the animals unable to walk or did they have a stiff-stilted gait?

Line 68. Although a detailed clinical examination was not possible, it should be possible to say something further about the gait just from observation. What about muscle mass? The animals were necropsied so this should have been noted? The sex of he animals should be noted. Were they all males? A serum CK activity is critical here. The animals were euthanized. It should have been possible to collect a blood sample just prior to death.

A figure showing muscle histopathology should be included.

Discussion.

The authors focus too much on hypokalemic polymyopathy when there is really no evidence for this disease. They should provide a differential for what this could be based on the pathological changes in the muscles. The authors stress the importance of identifying this condition in the lions but their approach is not likely to be fruitful. Start with the evidence you have. A better clinical description is critical here. Are these animals all male? What is the creatine kinase activity?

Are muscle samples archived on the animal that was necropsied? Immunostaining for dystrophy associated proteins (dystrophin, sarcoglycans and laminin alpha 2) sould be performed on the muscle specimens and a specific candidate may be identified from the stainings. Antibodies are commercially available that react with animal species (ie: canine and feline). This way a specific candidate gene can be identified.

I agree this is likely genetic.

Reference 11 is just an abstract. Is there a full paper with clinical and pathological details?

Round 2

Reviewer 1 Report

Thank you for your revisions to the original submission. The addition of clinical and post mortem findings and a greater discussion of differentials have, in my opinion, greatly improved the value and interest of the paper and I am happy now to recommend it for publication. 

Author Response

We are grateful to the reviewer for their comments and have carefully reviewed the document for grammatical and spelling errors.

Reviewer 3 Report

The authors have addressed some of my concerns but there are still a couple of points remaining.

Lines 84-93. The pathological changes do not rule out a form of muscular dystrophy. The X-linked dystrophin deficiency is less likely given the affected aniamals are both male and female. Many muscular dystrophies are autosomal recessive and these cannot be ruled out based on the pathology. Degeneration, regeneration and calcific deposits are classical findings in most species including domestic cats.

Line 268 further investigations. I agree that knowing if the affected animals are hypokalemic or not is an important point. While blood would be required to determine this, blood should also be evaluated for creatine kinase (CK) activity which would rule out many congenital myopathies if markedly elevated. This point should be added. In the end, whole exome or genome sequencing may be required to reach a diagnosis given the limitations to the study of these interesting animals.

Author Response

Thank you for your comments.

We have altered the wording (Lines 84-88) to indicate that muscular dystrophy could not be ruled out. 

We have also altered the wording (Lines 269-273) to indicate that creatinine kinase levels should be determined in affected cubs.